# Exploring the Influence of Context on Emotional Mimicry and Intention: An Affirmation of the Correction Hypothesis

**DOI:** 10.3390/bs13080677

**Published:** 2023-08-11

**Authors:** Xiaohui Xu, Ping Hu

**Affiliations:** Department of psychology, Renmin University of China, Beijing 100872, China; xuxiaohui@ruc.edu.cn

**Keywords:** emotional mimicry, context, electromyography (EMG), affiliative intention, distancing intention

## Abstract

Background: Emotional mimicry, a phenomenon frequently observed in our everyday interactions, is the act of replicating another individual’s facial expression. The Emotion Mimicry in Context View and the Correction Hypothesis underscore the critical role of context and intention within emotional mimicry. Methods: In two distinct studies, participants were presented with facial expressions of models (happiness and anger) within various contexts (affiliative, distancing, and neutral). Concurrently, we recorded electromyography (EMG) to index emotional mimicry, while participants explicitly rated the models’ intentions. Results: We observed context swiftly influences emotional mimicry within 500 ms, notably when the intentions of contexts are opposing to the intentions of facial expressions, leading to weakened muscle responses and diminished perceived intention. Furthermore, a notable correlation was discovered in the mimicry of angry faces; the more distancing the context, the stronger the corrugator supercilii (CS) muscle activity after context processing. Conclusions: First, emotional mimicry should not be simply viewed as an output corresponding to the expresser’s facial expressions but the dynamic process involving the active participation of the observer. Second, intention serves as a pivotal anchor, effectively integrating facial and contextual information. As such, we provided empirical support for the Correction Hypothesis.

## 1. Introduction

Emotional mimicry, defined as the mirroring of another individual’s facial expression, has been at the core of numerous scholarly investigations. The Emotion Mimicry in Context View, which encompasses a comprehensive review of previous studies, underscores that research on emotional mimicry often involves presenting participants with images or videos without any associated contextual information [1,2]. Context offers insights into the affiliative intentions between the observer and the expresser. For instance, the context could indicate whether the relationship shared by the individuals involved is cooperative or competitive [3], or if the observer perceives the expresser as part of their ingroup [4]. Consequently, the Emotion Mimicry in Context View emphasizes the critical role of context, asserting that emotional mimicry is more influenced by context than behavioral mimicry. This view suggests that emotional mimicry is most likely to occur within a context that is at minimum neutral, but preferably affiliative in nature [1,2].

To further clarify, two main concepts discussed above warrant more detailed examination. Firstly, the concept of ‘affiliation’ is pivotal. Fischer and Manstead recognized two distinct social goals in interpersonal interactions: affiliation, which is intended to cultivate positive social connections, and distancing, aimed at defining or preserving one’s social position vis-a-vis others and to protect one’s self-esteem, identity, or power, sometimes at the expense of others [5]. Emotions serve a vital function in fulfilling these social goals, where emotions like happiness, love, sadness, guilt, shame, and regret serve an affiliative function, while anger, hate, contempt, and disgust have a distancing function [5]. Taking into account Hess and Fischer’s proposition concerning the role of context in emotional mimicry, we can conceptualize affiliative and distancing intentions as context dimensions. The Emotion Mimicry in Context View, upon examining previous research on emotional mimicry, highlights the crucial role of affiliative intention [2]. For instance, individuals naturally mimic a smile more frequently than a frown, an action consistent with the affiliative orientation of happiness [1]. It has also been demonstrated that the likelihood of mimicry increases within an affiliative context. Examples of such contexts include mimicking the emotional expressions of those belonging to the same group [6,7], individuals characterized by positive attributes [8,9], and those engaged in cooperative behavior [3,10]. In these contexts, affiliative intention is often assumed as the default position [2,11]. However, the focus on intention implies that emotional mimicry occurs based on both the intention of the facial expression and the surrounding context. Predicting emotional mimicry becomes more complex when the intentions conveyed by the facial expression and the context are inconsistent, leading to mixed results under conditions such as the display of smiles from in-group members or frowns from out-group members. Consequently, these findings suggest that the concept of affiliative intention alone is insufficient to fully elucidate the nuances of emotional mimicry.

Secondly, ‘context’ is a multifaceted term that covers a wide range of information [12,13]. In the realm of emotional mimicry, context can incorporate environmental elements like social interactions; observer-specific characteristics or states such as empathy, attachment style, gender, and age; expresser-specific features like eye gaze; and the relationship between the observer and the expresser, such as group membership, familiarity, and cooperative or competitive relationships [14]. Moreover, visual or narrative stimuli presented alongside facial expressions can also be perceived as context [15,16]. The context has the potential to modify the interpretation of inherently affiliative expressions [1]. Hess and Hareli proposed that the adaptability of emotional responses depends on how easily any disparities between the appraisal of facial expressions and the context can be reconciled. If the context is congruent with the primary appraisal associated with the emotion, deriving meaning from the facial expression becomes an uncomplicated task. However, when a discordance occurs, individuals are prompted to consciously reassess this alignment, which could potentially lead to a reinterpretation of the emotional expression or a reappraisal of the context [13]. In parallel, the Correction Hypothesis within the realm of emotional mimicry recognizes the interplay between facial expressions and contexts [17]. It posits that the social modulation of emotional mimicry can be conceptualized as a corrective mechanism: when the context suggests that mimicry is inappropriate or undesired (i.e., it does not align with the observer’s affiliative aims), the observer attempts to counteract the impact of the observed expression on their own emotional display or feelings, resulting in a diminished or obstructed mimicry response. Therefore, the Correction Hypothesis tends to categorize “acontextual” congruent reactions to the expresser’s expressions as a form of uncorrected emotional mimicry. In contrast, it views muscle reactions to facial expressions influenced by the context as a type of “corrected” emotional mimicry.

Hence, emotional mimicry is neither a simple bottom-up nor top-down process, but a sophisticated response that contemplates both the stimuli and the interaction’s objective. Although the bottom-up and top-down processes of emotional mimicry have been investigated individually, the dual-process correction suggested by various theories is yet to be empirically substantiated [17,18]. Murata et al. devised an experimental paradigm where facial expressions, contextual introductions, and the same facial expressions are presented sequentially. During the contextual introduction, participants are specifically requested to infer the expresser’s emotion, e.g., “How does Hashimoto feel?” in order to incite the reflective mode, thereby leveraging significant cognitive resources [19]. This paradigm closely mirrors the correction process, where facial expressions displayed before and after the context independently highlight acontextual and contextual emotional mimicry, thereby emphasizing the impact of context.

As outlined above, there are three key points that require further elucidation in the understanding of emotional mimicry. Firstly, it remains uncertain whether affiliative contexts or contexts that align with facial intention foster emotional mimicry. While both the Emotion Mimicry in Context View and the Correction Hypothesis underscore the critical role of context in emotional mimicry, they present differing views. The Emotion Mimicry in Context View argues that affiliative contexts promote emotional mimicry. However, previous studies have found that affiliative contexts such as cooperation [3,10] and friendships [20] do not necessarily bolster the mimicry of angry expressions. On the other hand, the Correction Hypothesis suggests that emotional mimicry remains strong when the conveyed intention of the context matches the facial expression. Some studies have observed the mimicry of angry expressions in distancing contexts, such as frowning in response to outgroup members [21,22], or individuals exhibiting heightened states [23]. Therefore, empirical research is needed to further elucidate the role of context on emotional mimicry. Second, it remains unclear how observers discern the expresser’s intention when the information conveyed by the face and the context is inconsistent. Hess and Fischer proposed that context could reshape the social interpretation of inherently affiliative expressions [1]. The Correction Hypothesis places intention within its corrective process. However, while intention often serves as an explanatory mechanism in past research [4,6], it has never been directly measured. To address this gap, intention could be explicitly rated in future studies to provide evidence for the influence of context on intention. Thirdly, the relationship between intention and emotional mimicry calls for more empirical support. According to the Emotion Mimicry in Context View, the direction of intention, particularly affiliation, facilitates emotional mimicry. However, the Correction Hypothesis suggests that the alignment of intention between the facial expression and the context influences emotional mimicry. To date, few studies have explicitly explored the link between intention and emotional mimicry. Hess, for example, noticed a correlation between emotional mimicry and the quality of interaction, observing that mimicry enhanced interaction quality in an affiliative context but diminished it in a hostile one [24]. Therefore, direct evidence for the relationship between intention and emotional mimicry needs to be provided to further our understanding of these phenomena.

In conclusion, our research delves into the influence of context on emotional mimicry and intention, as well as the correlation between intention and emotional mimicry. In Study 1, we adopted Murata’s paradigm where participants sequentially viewed a facial expression, its context, and the same facial expression once more. We scrutinized the differences between acontextual and contextual emotional mimicry, thus discussing the effect of context on emotional mimicry as posited in the Correction Hypothesis. In Study 2, participants were simultaneously presented with the emotional expression and context, while three separate tasks were implemented to regulate their focus on face or context during mimicking emotions. This design further propelled our investigation into both the corrective outcomes and processes inherent in emotional mimicry.

## 2. Study 1

### 2.1. Materials and Methods

#### 2.1.1. Participants

Our study included a total of 34 undergraduate students, comprising 19 females and 15 males, with an average age of 20.44 ± 3.017 years. All participants had normal or corrected-to-normal vision and reported no history of neurological disorders. The study received ethical approval from the Institutional Review Board of the Department of Psychology. Participants were provided compensation for their participation upon the completion of the study.

#### 2.1.2. Stimuli

In Study 1, we used 12 video clips featuring Asian models (six females and six males) expressing happiness and anger as stimuli [25]. Each clip showed a model’s facial expression transitioning gradually from a neutral state to either happiness or anger over a duration of 1500 milliseconds. Regarding context, we crafted twelve sentences each to denote affiliative contexts (e.g., “I want to make friends with you”), distancing contexts (e.g., “I don’t want to make friends with you”), and neutral contexts (e.g., “What matters is the quality of a friend, not the quantity of friends”). These contexts were derived from the definitions of intention by Winter and Fischer et al. [5,26]. Participants were informed that these statements were attributed to the model, for instance, “The model is saying to you, ‘I want to make friends with you’.”

#### 2.1.3. Procedure

Participants were individually tested in a quiet laboratory room. After providing informed consent, they were seated comfortably in front of a monitor and electrodes were affixed to monitor muscle activity during the experiment.

The task used in this study was a modification of Murata’s task [19], as illustrated in Figure 1 of their study. Each trial began with a fixation cross-displayed for 1000 ms, followed by a 1500 ms video clip showing the facial expression. Subsequently, the social context was presented for a duration of 5000 ms. Then, participants were asked to rate the model’s intention on a scale ranging from 1 (indicating a complete distancing intention) to 9 (indicating a complete affiliative intention). The same video clip was replayed for an additional 1500 ms. Finally, a blank screen was displayed for a random interval of 4000–6000 ms to allow facial muscle activity to return to baseline levels [19,27,28]. The experiment consisted of two blocks, with each block randomly presenting one type of emotional expression. The order of the blocks was counterbalanced across participants to minimize any potential order effects.

#### 2.1.4. Facial EMG

The Biopac system (BIOPAC Systems, Inc., Santa Barbara, CA) was used to record the electromyographic (EMG) activity of each muscle at a sample rate of 1000 Hz. Consistent with previous studies, we used the activity of the zygomaticus major (ZM) to gauge smiles linked to happiness, and the activity of the corrugator supercilii (CS) to measure frowns associated with anger [19]. Prior to electrode placement, the skin was thoroughly cleaned with disinfectant alcohol and pumice gel (Skin Pure, Nihon Kohden) to ensure optimal signal transmission. All electrodes were Ag/AgCl miniature surface (EL254S, BIOPAC Systems, Santa Barbara, CA, USA) Electrodes for ZM and CS were positioned on the left side of the face, which has been shown to exhibit higher mimicry rates compared to the right side [29]. The reference electrode was placed on the mastoid process of the right ear [30].

For each trial, a response window was established, spanning 1000 ms before and 1500 ms after the stimulus onset. EMG data were processed using AcqKnowledge software version 5.0 (Biopac Systems). The data were pre-processed using a bandpass filter set between 20 and 500 Hz, with the root-mean-square (RMS) calculated, and EMG responses computed by subtracting the 1000 ms baseline from each subsequent 1500 ms interval [31]. To examine the indicators of emotional mimicry, we conducted a 2 (Emotion: Happiness, Anger) * 2 (Muscle: ZM, CS) repeated-measures ANOVA on muscle mean activities of both pre and post context. The results yielded significant interaction effects, *F* (1, 33) = 7.764, *p* = 0.009, η^2^ = 0.19. It suggests that the activity of the ZM muscle was higher and the activity of the CS muscle was lower when participants were exposed to happy faces compared to angry faces. Therefore, in the subsequent analysis, we utilized the activity of the ZM muscle as an indicator of mimicking happiness, and the activity of the CS muscle as an indicator of mimicking anger.

### 2.2. Results

#### 2.2.1. Examining the Influence of Context on Emotional Mimicry

For happy faces, we conducted a 3 (Context: Affiliative, Distancing, Neutral) * 2 (Process: Pre-context, Post-context) repeated-measures ANOVA on ZM activities (refer to Figure 2). The results revealed a nonsignificant main effect for context, *F* (2, 66) = 1.2, *p* = 0.308, η^2^ = 0.035, but a significant main effect for process, *F* (1, 33) = 8.095, *p* = 0.008, η^2^ = 0.197, and a marginally significant interaction effect, *F* (2, 66) = 2.791, *p* = 0.069, η^2^ = 0.078. Although participants exhibited less mimicry in response to happy faces after the context than before, pairwise comparisons revealed that the decline was significant after the distancing context (*p* = 0.009) and neutral context (*p* = 0.041) were presented. However, there was no significant difference between the muscle activities of happy faces before and after the presentation of the affiliative context (*p* = 0.900). Thus, in terms of contextual emotional mimicry, compared to acontextual emotional mimicry, the corrective mechanism may act to dampen mimicry when the context is either antagonistic or neutral.

For angry faces, we conducted a 3 (Context: Affiliative, Distancing, Neutral) * 2 (Process: Pre-context, Post-context) repeated-measures ANOVA on CS activities (refer to Figure 3). The results revealed a nonsignificant main effect for context, *F* (2, 66) = 0.373, *p* = 0.69, η^2^ = 0.011, and a nonsignificant interaction effect, *F* (2, 66) = 0.188, *p* = 0.829, η^2^ = 0.066. However, a significant main effect for process was observed, *F* (1, 33) = 4.608, *p* = 0.039, η^2^ = 0.123. This observation implies that participants demonstrated augmented muscular activity post-context-presentation as compared to prior, suggesting that context may impart supplementary information for angry faces, thereby fortifying emotional mimicry.

#### 2.2.2. Examining the Influence of Context on Intention

Firstly, we tabulated the trial numbers after classifying the intention ratings into three categories: affiliative intention (ratings from 1 to 4), distancing intention (ratings from 6 to 9), and neutral intention (rating of 5). It is noteworthy that we observed an equal number of trials for both affiliative intention and distancing intention when participants were presented with angry faces in an affiliative context (refer to Table 1 for specific details). This suggests that participants found it challenging to accurately judge the intention of the expresser under this particular condition. Furthermore, we performed two distinct 3 (Context: Affiliative, Distancing, Neutral) repeated-measures ANOVAs on intention ratings for both happy and angry faces. We observed significant main effects for happy faces, *F* (2, 66) = 108.226, *p* < 0.001, η^2^ = 0.871, and for angry faces, *F* (2, 66) = 123.591, *p* < 0.001, η^2^ = 0.885.

#### 2.2.3. Examining the Relationship between Intention and Emotional Mimicry

In our quest to examine the relationship between intention and mimicry, we calculated the corrective effect by subtracting pre-context EMG activities from post-context EMG activities. We proceeded to perform correlation analyses between intention and the corrective effect on ZM for happy faces (r = −0.266, *p* = 0.128), and between intention and the corrective effect on CS for angry faces (r = −0.316, *p* = 0.068). This implies that the stronger the affiliative intention in the context, the smaller the CS activities after the context.

## 3. Study 2

### 3.1. Materials and Methods

#### 3.1.1. Participants

A total of 44 students (35 females, 9 males; aged 21.18 ± 1.78 years) participated. All participants had either normal vision or vision corrected to normal, and none reported having any neurological disorders. The Department of Psychology’s Institutional Review Board approved the study. Upon completion of the experiment, participants received compensation for their time.

#### 3.1.2. Stimuli

We used eight video clips featuring Asian models (four females and four males) expressing happiness and anger as stimuli [25]. Each clip showed a model’s facial expression transitioning gradually from a neutral state to either happiness or anger over a duration of 3000 milliseconds. Like for Study 1, we constructed eight sentences to represent affiliative contexts, eight sentences for distancing contexts, and eight sentences for neutral contexts. Participants were informed that these statements were being made by the model, for example, “The model is saying to you, ‘I want to make friends with you’.”

We recruited 11 students (8 females, 3 males; aged 20.01 ± 1.49 years) to rate the level of intention expressed by these sentences on a scale from 1 (completely distancing intention) to 9 (completely affiliative intention), with 5 being neutral. A repeated-measures ANOVA revealed that the affiliative context (M ± SD = 7.44 ± 1.17) was rated as more affiliative than both the neutral context (M ± SD = 5.21 ± 0.75) and the distancing context (M ± SD = 2.11 ± 1.36, *F* (2, 11) = 753.049, *p* < 0.0001.

#### 3.1.3. Procedure

Participants were individually tested in a sound-attenuated laboratory room. They were seated comfortably in front of a monitor, approximately 50 cm away. After reading and signing an informed consent form, electrodes were affixed to their skin to monitor muscle activity during the experiment.

Participants were asked to imagine that they were engaged in a real-life conversation with the model displayed on the screen. As illustrated in Figure 4, prior to each block, instructions were displayed on the screen, informing participants of the task they were to complete in the upcoming block. There were three tasks: the face task, where participants were asked to focus their attention on facial expressions; the context task, where participants were asked to focus their attention on the context; and the face with context task, where participants were required to process information from both facial expressions and context. The order of blocks and tasks was counterbalanced across participants. Each trial began with the presentation of a fixation cross for 1000 ms. This was followed by a 3000 ms video clip, which depicted either happiness or anger, with a corresponding sentence displayed at the bottom of the screen. Subsequently, participants were asked to rate the degree of intention they felt towards the model (1 = completely distancing intention, 5 = neutral, 9 = completely affiliative intention). Finally, a blank screen was displayed for a random interval of 4000–6000 ms to allow facial muscle activity to return to baseline levels [19,27,28].

#### 3.1.4. Facial EMG

Similar to Study 1, the activity of the CS and ZM muscles was recorded using the Biopac system. The collected data then underwent pre-processing steps: bandpass filter (20–500 Hz), RMS calculation, and baseline correction (1000 ms pre-stimulus). To examine the indicators of emotional mimicry, we conducted a 2 (Emotion: Happiness, Anger) * 2 (Muscle: ZM, CS) repeated-measures ANOVA on muscle mean activities during 3000 ms. The results yielded significant interaction effect was observed, *F* (1, 43) = 7.266, *p* = 0.01, η^2^ = 0.145, suggesting that the activity of the ZM muscle was higher and the activity of the CS muscle was lower when participants were exposed to happy faces compared to angry faces. Therefore, in the subsequent analysis, we utilized the activity of the ZM muscle as an indicator of mimicking happiness, and the activity of the CS muscle as an indicator of mimicking anger.

### 3.2. Results

#### 3.2.1. Examining the Influence of Context on Emotional Mimicry

For happy faces, we performed a 3 (Context: Affiliative, Distancing, Neutral) * 3 (Task: Face, Context, Face with Context) repeated-measures ANOVA on ZM activities (see Figure 5). The results revealed nonsignificant main effects for both context, *F* (2, 86) = 0.75, *p* = 0.476, η^2^ = 0.017, and task, *F* (2, 86) = 1.312, *p* = 0.275, η^2^ = 0.03. However, a significant interaction effect was observed, *F* (4, 172) = 3.30, *p* = 0.012, η^2^ = 0.071. This indicates that when participants were presented with happy faces accompanied by a distancing context, the activity of the ZM muscle significantly decreased in the face with context task (M ± SD = −0.502 ± 0.292) compared to the face task (M ± SD = 0.963 ± 0.505), *p* = 0.004.

To further delineate the time course of emotional mimicry differences under various tasks within the distancing context, we divided the muscle activities of 3000 ms into six segments of 500 ms each. Repeated ANOVA was employed at each time segment to calculate the difference among different tasks. As illustrated in Figure 6, we observed the trends of electromyographic responses to happy faces: when participants focused their attention on the faces, we noted activation of the ZM muscle, indicating imitation of the happy faces. However, when participants processed information from a distancing context, the ZM muscle activity was at or near baseline, implying that the mimicry of happy faces was suppressed. Moreover, significant task differences were observed in the following time segments. Specifically, within 0–500 ms, *F* (2, 86) = 4.433, *p* = 0.015, η^2^ = 0.093. Pairwise comparisons highlighted a significant difference between the “face with context” task and the “context” task, *p* = 0.033 (Bonferroni-corrected). Within 500–1000 ms, *F* (2, 86) = 3.894, *p* = 0.024, η^2^ = 0.083. Pairwise comparisons pointed out a significant difference between the “face with context” task and the “face” task, *p* = 0.039 (Bonferroni-corrected). Within 1000–1500 ms, *F* (2, 86) = 4.941, *p* = 0.009, η^2^ = 0.103. Pairwise comparisons revealed significant differences between the “face with context” task and the “face” task, *p* = 0.017 (Bonferroni-corrected), as well as between the “face with context” task and the “context” task, *p* = 0.021 (Bonferroni-corrected). Within 1500–2000 ms, *F* (2, 86) = 4.624, *p* = 0.012, η^2^ = 0.097. Pairwise comparisons indicated significant differences between the “face with context” task and the “face” task, *p* = 0.029 (Bonferroni-corrected), and between the “face with context” task and the “context” task, *p* = 0.030 (Bonferroni-corrected). The results under other conditions can be found in Appendix A.

For angry faces, we conducted a 3 (Context: Affiliative, Distancing, Neutral) * 3 (Task: Face, Context, Face with Context) repeated-measures ANOVA on CS activities. As depicted in Figure 7, the results revealed nonsignificant main effects for both context, *F* (2, 86) = 0.922, *p* = 0.402, η^2^ = 0.021, and task, *F* (2, 86) = 1.984, *p* = 0.144, η^2^ = 0.044. Additionally, a nonsignificant interaction effect was observed, *F* (4, 172) = 1.099, *p* = 0.359, η^2^ = 0.025. However, the pairwise comparisons indicated that when participants were shown angry faces with an affiliative context, CS activity was significantly decreased in the face with context task (M ± SD = −0.166 ± 0.212) compared to the face task (M ± SD = 0.345 ± 0.199), *p* = 0.028.

In parallel to the analysis on happiness, the 3000 ms muscle activities to angry faces were segmented into six 500 ms sections, and a repeated ANOVA was performed at each interval to calculate disparities among tasks. As shown in Figure 8, we observed that when participants focused on the faces, activation of the CS muscle was noted, indicative of imitation of the angry faces. However, once participants processed information from an affiliative context, the muscle activity was at or near baseline, suggesting suppression of mimicry of the angry faces. Significant task differences were noted in three specific time intervals. Specifically, within the 0–500 ms range, *F* (2, 86) = 3.598, *p* = 0.032, η^2^ = 0.077. Pairwise comparisons revealed a significant difference between the “face with context” task and the “context” task, *p* = 0.021, Bonferroni-corrected. In the 1500–2000 ms interval, *F* (2, 86) = 3.333, *p* = 0.04, η^2^ = 0.072. Pairwise comparisons demonstrated a significant difference between the “face with context” task and the “face” task, *p* = 0.045, Bonferroni-corrected. Within the 2000–2500 ms time span, *F* (2, 86) = 3.118, *p* = 0.049, η^2^ = 0.068. Pairwise comparisons indicated a marginally significant difference between the “face with context” task and the “face” task, *p* = 0.07, Bonferroni-corrected. The results under other conditions can be found in Appendix A.

#### 3.2.2. Examining the Influence of Context on Intention

As in Study 1, we categorized the intention ratings into three groups and tallied the number of trials for each (refer to Table 2 for specifics). We observed that participants struggled to accurately discern the intention of the expresser displaying angry faces within an affiliative context. Moreover, we conducted two separate 3 (Context: Affiliative, Distancing, Neutral) * 3 (Task: Face, Context, Face with Context) repeated-measures ANOVAs on ratings of intention for happy faces and angry faces.

For happy faces, significant main effects were observed for both context, *F* (2, 86) = 374.742, *p* < 0.001, η^2^ = 0.897, and task, *F* (2, 86) = 112.104, *p* < 0.001, η^2^ = 0.723. Participants rated happy faces with an affiliative context (M ± SD = 7.207 ± 0.114) higher than those with a neutral context (M ± SD = 6.252 ± 0.083) and a distancing context (M ± SD = 4.453 ± 0.062). Similarly, happy faces were rated higher under the face task (M ± SD = 6.785 ± 0.119) than under the face with context task (M ± SD = 5.849 ± 0.079) and the context task (M ± SD = 5.277 ± 0.058). A significant interaction effect was also observed, *F* (4, 172) = 175.544, *p* < 0.001, η^2^ = 0.803. This suggests that, when participants were shown happy faces within a distancing or neutral context, intention significantly decreased in the face with context task and the context task compared to the face task. Conversely, when participants were shown happy faces within an affiliative context, intention significantly increased in the face with context task and the context task compared to the face task.

For angry faces, the results revealed significant main effects for both context, *F* (2, 86) = 261.67, *p* < 0.001, η^2^ = 0.859, and task, *F* (2, 86) = 113.512, *p* < 0.001, η^2^ = 0.756. Participants rated angry faces with an affiliative context (M ± SD = 5.262 ± 0.072) higher than those with a neutral context (M ± SD = 4.458 ± 0.067) and a distancing context (M ± SD = 3.028 ± 0.109). Similarly, participants rated angry faces under the face task (M ± SD = 3.474 ± 0.104) lower than under face with context task (M ± SD = 4.211 ± 0.088) and the context task (M ± SD = 5.063 ± 0.053). Moreover, a significant interaction effect was observed, *F* (4, 172) = 168.259, *p* < 0.001, η^2^ = 0.796. This suggests that when participants were shown angry faces within an affiliative or neutral context, intention significantly increased in the face with context task and the context task compared to the face task. Conversely, when participants were shown angry faces within a distancing context, intention significantly decreased in the face with context task and the context task compared to the face task.

#### 3.2.3. Examining the Relationship between Intention and Emotional Mimicry

To investigate the direct relationship between intention and emotional mimicry, we conducted correlation analyses on intention ratings and ZM activity for happy faces, as well as intention ratings and CS activity for angry faces, separately for each of the three tasks (refer to Table 3 for details). Our results showed significant correlations among the intention ratings and among the muscle activities across the three tasks. However, we found almost no significant correlation between intention and muscle activities, except for a noteworthy exception. Specifically, we observed a significant negative correlation between intention ratings in the context task and CS activities in the face with context task (r = −0.302, *p* = 0.046). This implies that a stronger affiliative intention in the context was associated with weaker CS activities in response to angry faces under the face with context task.

## 4. Discussion

In the present study, through two experiments, we investigated the influence of context on emotional mimicry and intention, as well as their interconnection. Firstly, we observed both uncorrected and corrected emotional mimicry. Contexts with intentions opposing facial expressions (e.g., distancing context to happy face, affiliative context to angry face) play crucial roles in corrected emotional mimicry. On the one hand, these contexts weaken the muscle response to the facial expression, and on the other hand, they diminish the intention conveyed by the face. Furthermore, we noted that the correction occurs very early, within approximately 500 ms. Secondly, we found that the relationship between intention and emotional mimicry does not depend on a specific direction (e.g., affiliative), but rather, they should align with each other. For example, we observed that the more distancing the context, the stronger the CS muscle activity was for angry faces after context processing. From these findings, we derive the following conclusions: First, we propose a redefinition of emotional mimicry. Rather than viewing it as an outcome that aligns with the expresser’s facial expressions from the observer’s perspective, we argue it should be seen as a dynamic process of facial muscle activity resulting from the observer’s processing of facial expressions and contexts. This process necessitates conscious participation and requires the integration of information from both facial expressions and contexts. This perspective is in line with both the Emotion Mimicry in Context View and the Correction Hypothesis. Second, we propose that intention does not merely act as a facilitator in emotional mimicry when it is affiliative, but it functions as a tether that integrates facial and contextual information. In situations where information is consistent, emotional mimicry does not require correction, but when the information is conflicting, it does. In this respect, our perspective deviates from the Emotion Mimicry in Context View and leans more towards the Correction Hypothesis.

Both the Emotion Mimicry in Context View and the Correction Hypothesis underscore the integral role of context in emotional mimicry. The former distinguishes emotional mimicry from mere behavioral imitation, while the latter separates automatic (bottom-up) from more deliberate (top-down) emotional mimicry. Our current research supports a dual-process model of emotional mimicry. More specifically, we found evidence of automatic mimicry for happy faces in a context-free setting in Study 1, and for both happy and angry faces when participants were directed to concentrate solely on facial expressions in Study 2. However, when participants encountered an opposing context, we recorded the correction in emotional mimicry. This observation was consistent across both experiments. Hence, results in the present study fit within the framework of dual-process theories, which incorporate bottom-up emotional mimicry [32,33] and top-down emotional mimicry [4,6].

From this perspective, emotional mimicry should be perceived as a process rather than a mere outcome; otherwise, we risk overlooking valuable information. Notably, in our study, we found that the corrective process occurs within 500 ms. Indeed, numerous studies have investigated the effect of emotional context on facial emotion processing, indicating that emotional context significantly modulates early ERP amplitudes evoked by emotionally expressive faces [34,35,36]. The N400 component, which is considered a reflection of the influence of emotional context during information integration, has received significant attention [35,36]. Crucially, the N400 is a negative deflection that shows amplified amplitude when incongruent stimuli are presented. This corresponds with our findings that within 500 ms of the concurrent presentation of a facial expression and a context that contradicts the implied intention. This impact was even evident in the electromyographic responses within 500 ms, suggesting that participants may have accomplished the integration of emotional facial and contextual information, thereby resulting in an apt facial response. While previous studies have utilized EEG to investigate emotional mimicry [37,38] they have not encompassed the conditions of integrating facial emotions and context. Thus, our research provides substantial evidence for the outcome of integrating emotional facial and contextual information in emotional mimicry, paving the way for further exploration into this integration process. Future research may leverage tools like EEG to uncover the neural mechanisms underpinning the process of emotional mimicry and pinpoint the brain activity most closely associated with ultimate behavioral expressions.

While the Emotion Mimicry in Context View underscores the crucial role of affiliative intention in emotional mimicry, our findings suggest the key factor is not the direction of the intention, but the consistency between the intentions expressed by the facial expressions and the context. Indeed, previous studies have presented mixed results under conditions where facial expression and context are not matched. For instance, some studies have discovered that individuals mimic the smiles of outgroup members, frowns of ingroup members [4,6,7,21,22], and happy expressions of individuals in high states [23]. However, other studies have failed to observe emotional mimicry when participants were exposed to happy expressions of competitors or angry expressions of cooperators [3,10], smiles from strangers or frowns from friends [20], and angry faces of positive individuals and happy faces of negative individuals [9]. Likewise, in the present study, we observed that mimicry for angry faces is not enhanced when the context is affiliative. Upon the reclassification of intention ratings into three categories, participants encountered difficulties accurately judging the expresser’s intention in an affiliative context with an angry face. In addition, correlation analyses from both experiments showed that mimicry of anger increases with more distancing context, not affiliative. This aligns with previous studies that demonstrate that anger mimicry does not necessarily take place even in affiliative settings [39,40]. Thus, an overemphasis on the affiliative nature of intention might overshadow the positive role that distancing intention can play. For example, Rauchbauer et al. posited that frowning in response to outgroup members could serve to pacify a perceived threatening interaction partner with the goal of defusing a potentially adverse social exchange [21]. Mauersberger et al. found that individuals with superior emotional regulation skills displayed congruent anger, facilitating positive social interactions [8]. This aligns with the notion that appropriate anger expression can be part of socially adaptive responses [41] and can yield long-term benefits [42]. Consequently, we propose that the adaptability of emotional mimicry is extensive, not just confined to promoting affiliative relationships but also mitigating hostile ones.

Our research provides empirical support for the Correction Hypothesis. We believe this study could also inspire insights in other fields. Essentially, this research investigates the influence of multimodal emotional information on the participant’s emotional mimicry. In the realm of human–computer interaction, computers often present users with multimodal emotional information as well, such as visually displayed text, audibly presented voice information, and even tactile sensations. Understanding how users perceive and respond to this multimodal emotional information can help us optimize products more effectively. Nonetheless, this study does have some limitations. For instance, we categorized happiness as an affiliative emotion and anger as a distancing emotion based on the social function of emotions [5]. We manipulated affiliative, distancing, and neutral contexts based on the definitions of affiliation and distancing. However, it is important to recognize that there may be confounding factors such as valence and arousal at play. For instance, happy faces and distancing contexts are not only conflicting in intention but also in valence. Although we attempted to direct participants’ attention towards intention through instructions and question setting in this study, we cannot guarantee that this method fully isolated the influence of valence. Thus, future research needs to implement more refined research designs to disentangle intention from valence and other confounding factors, thus providing more explicit empirical support for the Correction Hypothesis. Additionally, this study focused solely on happiness and anger as the subjects of emotional mimicry and found a relatively stable correction effect only on happiness. Therefore, in future research, it is necessary to further investigate whether anger follows the correction process and to explore whether other emotions, such as sadness and fear, exhibit unique behaviors in the correction process.

## 5. Conclusions

In summary, our study demonstrates that context swiftly influences emotional mimicry within 500 ms, notably when intentions oppose facial expressions, leading to weakened muscle responses and diminished perceived intention. The interplay between intention and emotional mimicry is not direction-specific but is more about alignment. We argue for a redefinition of emotional mimicry as a dynamic process involving the observer’s active integration of facial and contextual information. Furthermore, intention serves as an anchoring element in this integrative process. Collectively, our findings offer robust empirical support for the Correction Hypothesis.

## Figures and Tables

**Figure 1 behavsci-13-00677-f001:**
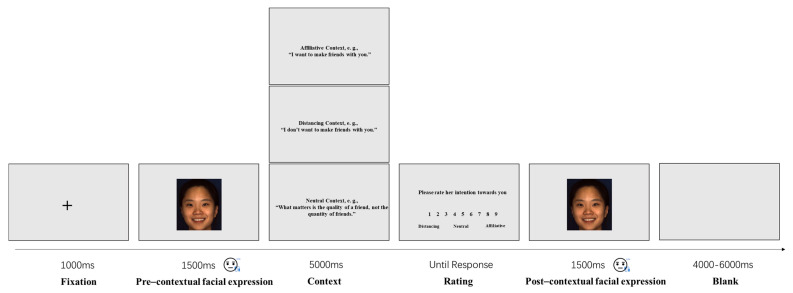
Illustration of the timeline of one trial in Study 1. Picture used with permission [25], ©2008, IEEE.

**Figure 2 behavsci-13-00677-f002:**
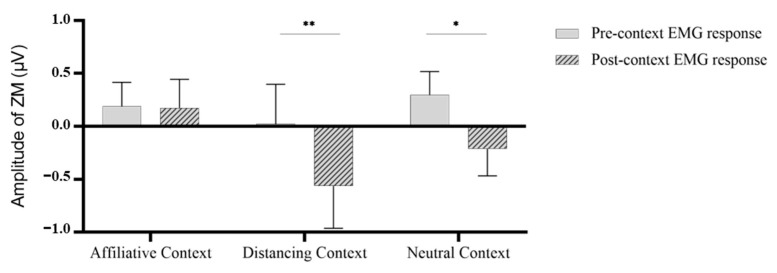
Mean pre- and post-context activities in zygomaticus major to happy faces. *, *p* < 0.05, **, *p* < 0.01.

**Figure 3 behavsci-13-00677-f003:**
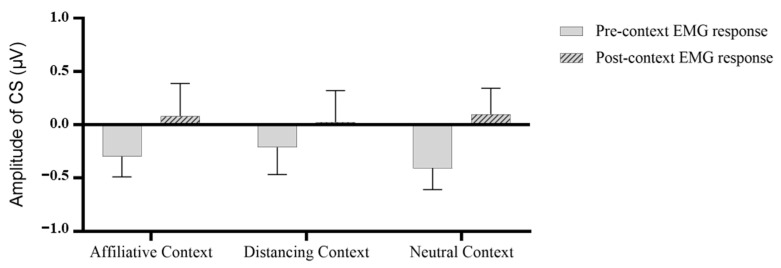
Mean pre- and post-context activities in corrugator supercilii to angry faces.

**Figure 4 behavsci-13-00677-f004:**
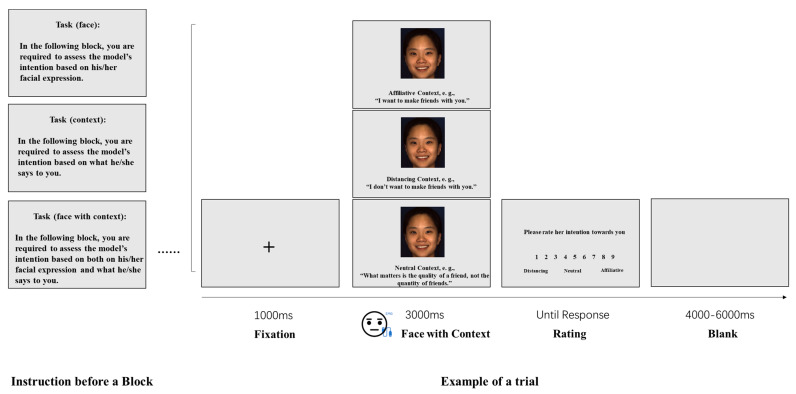
Illustration of the timeline of one trial in Study 2. Picture used with permission [25], ©2008, IEEE.

**Figure 5 behavsci-13-00677-f005:**
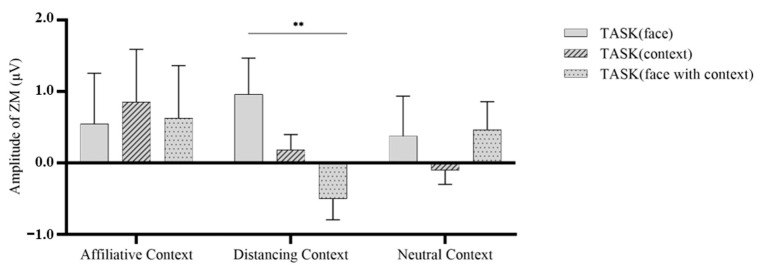
Mean zygomaticus major activity in response to happy faces across tasks. **, *p* < 0.01.

**Figure 6 behavsci-13-00677-f006:**
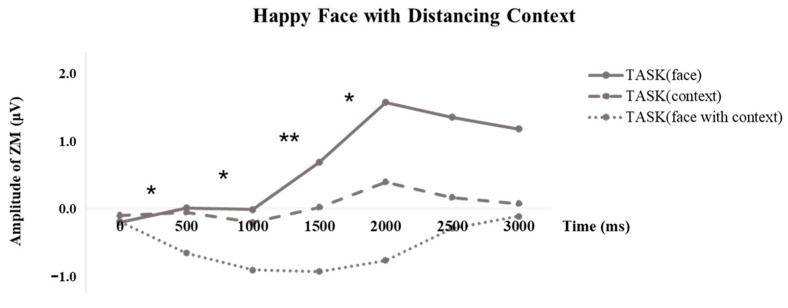
Timeline of zygomaticus major activity to happy faces with distancing context. *, *p* < 0.05, **, *p* < 0.01.

**Figure 7 behavsci-13-00677-f007:**
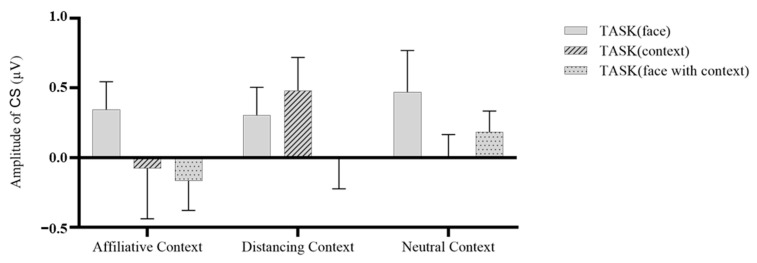
Mean corrugator supercilii activity in response to angry faces across tasks.

**Figure 8 behavsci-13-00677-f008:**
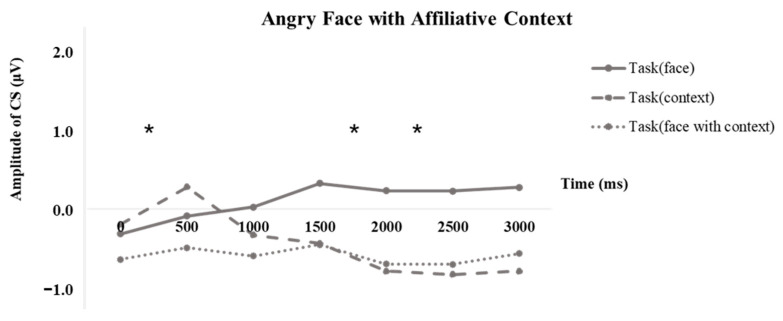
Timeline of corrugator supercilii activity to angry faces with affiliative context. *, *p* < 0.05.

**Table 1 behavsci-13-00677-t001:** Mean values and trial counts for intention rating.

Variables		Mean ± SD	Number of Trials: Rating as Distancing Intention (1–4)	Number of Trials: Rating as Affiliative Intention (6–9)	Number of Trials: Rating as Neutral Intention (5)
Happiness	Affiliative	7.04 ± 0.90	22	364	22
	Neutral	5.83 ± 0.53	34	235	139
	Distancing	3.03 ± 0.96	347	29	32
Anger	Affiliative	5.11 ± 1.28	168	191	49
	Neutral	4.45 ± 0.75	172	56	180
	Distancing	2.45 ± 0.78	382	14	12

**Table 2 behavsci-13-00677-t002:** Mean values and trial counts for intention rating.

Variables		Happiness			Anger		
		Affiliative	Neutral	Distancing	Affiliative	Neutral	Distancing
Task (face)	Mean ± SD	3.483 ± 0.737	3.52 ± 0.701	3.42 ± 0.729	6.884 ± 0.793	6.818 ± 0.775	6.653 ± 0.894
	Number of Trials: Rating as Distancing Intention (1–4)	15	16	26	279	286	296
	Number of Trials: Rating as Affiliative Intention (6–9)	317	315	300	31	30	27
	Number of Trials: Rating as Neutral Intention (5)	20	21	26	42	36	29
Task (context)	Mean ± SD	7.114 ± 0.881	5.293 ± 0.45	2.784 ± 0.86	7.352 ± 0.88	5.588 ± 0.624	2.892 ± 0.87
	Number of Trials: Rating as Distancing Intention (1–4)	3	16	309	11	34	321
	Number of Trials: Rating as Affiliative Intention (6–9)	334	142	17	324	107	10
	Trial Number: Rating as Neutral Intention (5)	15	194	26	17	211	21
Task (face with context)	Mean ± SD	5.19 ± 0.922	4.563 ± 0.664	2.881 ± 0.858	7.386 ± 0.809	6.349 ± 0.688	3.813 ± 0.848
	Number of Trials: Rating as Distancing Intention (1–4)	6	14	243	138	170	303
	Number of Trials: Rating as Affiliative Intention (6–9)	331	272	55	165	62	26
	Number of Trials: Rating as Neutral Intention (5)	15	66	54	49	120	23

**Table 3 behavsci-13-00677-t003:** Correlation between intention and ZM activities of happiness and CS activities of anger.

Correlation between Intention and ZM Activities of Happiness						
Variables	1	2	3	4	5	6
1. intention in TASK (face)	1					
2. ZM in TASK (face)	−0.049	1				
3. intention in TASK (context)	0.102	0.019	1			
4. ZM in TASK (context)	−0.040	0.855 **	0.107	1		
5. intention in TASK (face with context)	0.575 **	−0.094	0.440 **	−0.026	1	
6. ZM in TASK (face with context)	−0.031	0.883 **	0.190	0.900 **	−0.054	1
Correlation between Intention and CS Activities of Anger						
Variables	1	2	3	4	5	6
1. intention in TASK (face)	1					
2. CS in TASK (face)	−0.252	1				
3. intention in TASK (context)	0.023	−0.107	1			
4. CS in TASK (context)	0.081	0.308 *	0.044	1		
5. intention in TASK (face with context)	0.683 **	−0.086	0.263	0.083	1	
6. CS in TASK (face with context)	−0.077	0.714 **	−0.302 *	0.424 **	−0.115	1

ZM means zygomaticus major, CS means corrugator supercilia. *, *p* < 0.05, **, *p* < 0.01.

## Data Availability

The data presented in this study are available on request from the corresponding author.

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
