# Peer review of "Exploring the Influence of Context on Emotional Mimicry and Intention: An Affirmation of the Correction Hypothesis"

_behavsci, 2023, doi:10.3390/bs13080677_

Round 1

Reviewer 1 Report

Dear Autor(s). Please review the suggestions made to increase the quality of the article.

It is good research. Despite this, the manuscript includes a lot of redundant information in some paragraphs. Reviewing this content and including a summary with the most relevant data is suggested. In addition, please, review the following suggestions about this work:

·      Revise the content of the Abstract as it does not conform to the four items recommended by the journal (Background, Methods, Results, and Conclusion).

·      Adjust the number of words in the abstract to the journal's standard of 200 words.

·      The document is not well cited and referenced according to the MDPI journal standard.

·      Revise the literature review in the introduction as there is information that should be included in the discussion rather than under this heading, for example, lines 79-91.

·      I believe that a single study would have been adequate as the inclusion of a second study tends to make the information more extensive and the comparison a little confusing, so it is suggested that the two studies be compiled with the most important data from each to avoid repeating the information and to make a single discussion of the two.

·      Therefore, in the discussion section, this should be a single one for both studies, and the discussion should be sufficiently supported and justified by references from other authors.

·      As it is a very long discussion, it is suggested that it be revised to include conclusions of the research, as the journal's standard mentions: Conclusions: This section is not mandatory but can be added to the manuscript if the discussion is unusually long or complex. As is the case with your manuscript.

·      Check references according to journal standards.

If there is some form of compensation during an experiment, the data tends to skew the results ¿Was this factor considered with the experimental study groups?

Author Response

Response to Reviewer 1 Comments

Point 1: Revise the content of the Abstract as it does not conform to the four items recommended by the journal (Background, Methods, Results, and Conclusion). Adjust the number of words in the abstract to the journal's standard of 200 words.

Response 1: We are grateful for your valuable feedback. Following your recommendations, we've revised our abstract within 200 words, distinctly categorizing it under the sections of Background, Methods, Results, and Conclusion.

Point 2: The document is not well cited and referenced according to the MDPI journal standard.

Response 2: We sincerely appreciate your recommendations. As a result, we have adjusted and updated our citation style accordingly.

Point 3: Revise the literature review in the introduction as there is information that should be included in the discussion rather than under this heading, for example, lines 79-91.

Response 3: We greatly value your suggestions. We have completely rewritten the introduction to ensure brevity and a smoother flow of logic. Specifically, we initially introduced two critical concepts derived from the Emotion Mimicry in Context View: affiliation and context. We then separately discussed concepts and studies related to affiliation and context, which naturally led to the introduction of the Correction Hypothesis. Based on the identified gaps in the existing research, we proposed three issues that still need to be empirically tested. Finally, we provided a summary. As for the literature you mentioned in the original lines 79-91, we have moved them to the discussion section, specifically lines 485-494.

Point 4: I believe that a single study would have been adequate as the inclusion of a second study tends to make the information more extensive and the comparison a little confusing, so it is suggested that the two studies be compiled with the most important data from each to avoid repeating the information and to make a single discussion of the two. Therefore, in the discussion section, this should be a single one for both studies, and the discussion should be sufficiently supported and justified by references from other authors.

Response 4: We sincerely appreciate your suggestions, as they have been instrumental in our article revision process.

Firstly, we revisited the results of our two experiments, further determining that our presentation of results should correspond to the research questions proposed, presented in the following order: (1) the influence of context on emotional mimicry; (2) the influence of context on intention; (3) the relationship between emotional mimicry and intention. Upon reviewing our original paper, we did indeed find some results that could potentially cause a lack of clarity in the narrative logic.

To begin with, in the original version, we presented the calculation of the electromyography indicators of emotional mimicry as a part of results (lines 264-276), which might have confused the reader. Thus, in the current version, we moved this section under the methods of electromyography (lines 192-199 in study 1, lines 294-302 in study 2). We still explained why we chose the corrugator supercilii muscle as an indicator of anger mimicry and the zygomaticus major muscle for happiness mimicry, but this no longer occupies space in the results section, thus making the results clearer.

Secondly, in the original version, we also conducted comparisons of face and context effects (lines 303-317, lines 360-368). But in the current version, we decided to remove this part as we felt it had a weaker association with the core issues.

In addition, based on Reviewer 3's suggestion, we have added a description and analysis of the process of emotional mimicry (lines 315-338, lines 349-365). We believe these results can provide richer information about the correction process in emotional mimicry. Indeed, we found that emotional mimicry correction may occur within 500ms after stimulus (simultaneous presentation of faces and situations).

Secondly, it should be noted that we have decided to keep both experiments. However, for the sake of logical narration, we swapped the original study 1 and 2. In study 1, we used contextual and acontextual emotional mimicry to highlight the role of context. In study 2, we presented context and faces simultaneously, but manipulated participants' attention to explore the role of context.

Thirdly, we have omitted the specific discussions pertaining to Study 1 and Study 2, reconfiguring our discussion section to foster a more comprehensive dialogue with existing research in the field. Moreover, we have also incorporated sections contrasting our study with previous research. For example, in lines 461-473, we have drawn from past ERP studies to discuss the timing of corrections, and in lines 487-512, we have used previous research on anger mimicry to discuss the intention matching aspect of emotional mimicry.

Point 5: As it is a very long discussion, it is suggested that it be revised to include conclusions of the research, as the journal's standard mentions: Conclusions: This section is not mandatory but can be added to the manuscript if the discussion is unusually long or complex. As is the case with your manuscript. Check references according to journal standards.

Response 5: We sincerely appreciate your suggestions. We have added a conclusion section (lines 540-549).

Point 6: If there is some form of compensation during an experiment, the data tends to skew the results? Was this factor considered with the experimental study groups?

Response 6: We appreciate your suggestion. In fact, we believe that the compensation offered to participants is unrelated to the experimental tasks and would not affect the experiment results. When explaining the experiment to participants, we clarify two points: First, our task requires participants to observe the model's expressions and context information and respond to the model. This response is not about right or wrong and will not affect the final distribution of compensation. Second, the compensation given after the experiment is a token of appreciation for participants' involvement in the experiment and has no relationship with the characters in the experiment.

Reviewer 2 Report

Dear Authors,

I have had the opportunity to review your article, and I must say that your study is very promising. Your paper is comprehensive and provides a good explanation of the topic.

I am curious to know more about how your research can be used in different sectors. From my understanding, your approach could potentially be beneficial for various research fields such as behavior analysis, psychology, neuromarketing, and clinical studies. It would be interesting to know how your findings could complement the research in other fields and how it could improve the accuracy and efficiency of physiological data collection.

Some individuals could lack expressivity; in this sense, could your study be complemented with other physiological methods for emotion analysis? For example, ECG or EEG?

While the concepts you've presented are compelling, I believe that providing concrete examples of future applications would significantly enhance the strength of your paper. Please elaborate on this in your discussion section.

I hope that my feedback will be helpful in further improving your study.

Author Response

Response to Reviewer 2 Comments

Point 1: I am curious to know more about how your research can be used in different sectors. From my understanding, your approach could potentially be beneficial for various research fields such as behavior analysis, psychology, neuromarketing, and clinical studies. It would be interesting to know how your findings could complement the research in other fields and how it could improve the accuracy and efficiency of physiological data collection. Some individuals could lack expressivity; in this sense, could your study be complemented with other physiological methods for emotion analysis? For example, ECG or EEG? While the concepts you've presented are compelling, I believe that providing concrete examples of future applications would significantly enhance the strength of your paper. Please elaborate on this in your discussion section.

Response 1: We are honored to receive your response and appreciate your valuable suggestions. According to your advice, we have made the following modifications: from line 473 to 481, based on our current research results, we proposed future research directions involving the use of EEG. From line 513 to 521, considering the nature of our current study, we proposed potential research directions in the field of human-computer interaction.

Reviewer 3 Report

The paper presents experimental results demonstrating the effects of the Correction hypothesis and verifying the strong relation between context, intention and emotional mimicry.

It is very well writen and reading goes smoothly. The literature survey fulfills the requirements for presentation of previous work and information.

Some points for consideration:

1. The values in Table 1 and Table 2 refer to the activity of ZM and CS recorded by the EMG device. However, it is not clear what these numbers refer to (mean milliVolts ?) and for what period of time.

2. Additional to comment #1, it would be very helpful to provide a graph of how the activation of these two muscles varies over time, especially when the correction is activated where one would expect a sudden change in the recorded values. It is assumed from the rest of the analysis but it would add more to the description of the procedures and the explanation.

3. Moreover, the mean time after the correction event (the mismatching intetion) occured when muscle activity starts to change would be very informative. This could then be related to the current and general mood of the observer, etc.

Author Response

Response to Reviewer 3 Comments

Point 1: The values in Table 1 and Table 2 refer to the activity of ZM and CS recorded by the EMG device. However, it is not clear what these numbers refer to (mean milliVolts ?) and for what period of time.

Response 1: We sincerely appreciate your suggestions. In the original draft, the units of electromyographic activity in Table 1 and Table 2 were both microvolts, representing the average within 3000ms of emotional facial presentation.

However, it's worth mentioning that in the original version, we introduced the calculation of electromyographic indicators of emotional mimicry as part of the results section (lines 264-276, where Table 1 is located). This could potentially confuse readers. Therefore, in the current version, we've moved this section to fall under the methods section discussing electromyography (lines 192-199 in Study 1, lines 294-302 in Study 2). We still elaborate on our choice of the corrugator supercilii muscle as an indicator of anger mimicry and the zygomaticus major muscle for happiness mimicry. However, this no longer takes up space in the results section, leading to clearer results. Secondly, in the original manuscript, we also conducted comparisons of face and context effects (lines 303-317, lines 360-368, where Table 2 is located). However, in the current version, we've decided to omit this part as it seems to have a weaker association with the central research questions.

Point 2: Additional to comment #1, it would be very helpful to provide a graph of how the activation of these two muscles varies over time, especially when the correction is activated where one would expect a sudden change in the recorded values. It is assumed from the rest of the analysis but it would add more to the description of the procedures and the explanation.

Response 2: We are grateful for your suggestions, which are immensely helpful for presenting our results. However, it's important to note that, following the advice of Reviewer 1, we've reconsidered our structure and swapped the order of the original Studies 1 and 2. Therefore, in the current version of Study 2, we segmented the data into 500ms intervals to depict the process of emotional mimicry over time (line 337, line 364).

It's worth mentioning that, in fact, we created segmented graphs for all conditions in both studies. However, in Study 1, due to the order of stimulus presentation being emotional face, context, and then the same emotional face again, it's not suitable to depict the difference between emotional mimicry with and without context over time. Therefore, we didn't add related content under Study 1. Secondly, in Study 2, although we drew graphs of timeline for all conditions, presenting them all might lead to redundancy. Thus, we only displayed the timeline for conditions that showed significant differences.

Point 3: Moreover, the mean time after the correction event (the mismatching intetion) occured when muscle activity starts to change would be very informative. This could then be related to the current and general mood of the observer, etc.

Response 3: We deeply appreciate your suggestions, which have been very helpful in presenting our results. As mentioned in our response to your second suggestion, we segmented the data into 500ms intervals to illustrate the process of emotional mimicry over time. For each interval, we performed a repeated-measures ANOVA for different tasks (face, context, face with context) to help identify when the correction occurs (lines 315-336, 349-363). The results revealed that the correction occurred as early as 500ms when the emotional face and context were presented simultaneously. In the discussion section, we further associated this finding with the cognitive mechanisms of emotion and context integration (lines 487-512). In addition, we included the results of the repeated-measures ANOVA for each condition in Appendix A (line 567).

Round 2

Reviewer 1 Report

The new version of the paper shows that the suggestions made have been considered, and the manuscript has been substantially improved and is therefore recommended for publication.

However, it is worth mentioning that there are a couple of points that the authors should consider:

- Under heading 4 Discussion. It is suggested to start the sentence with another connector as it may confuse the conclusions.

- Line 524. Keep the same outline as the previous quotations.

- Check lack of space line 475.

- Check Reference 5. It is incomplete.

Best regards.

Author Response

Response to Reviewer 1 Comments

Point 1: Under heading 4 Discussion. It is suggested to start the sentence with another connector as it may confuse the conclusions.

Response 1: We appreciate your valuable input. In response to your suggestions, we've edited the connector at line 417. The revised version reads, " In the present study, through two experiments, we investigated the influence of context on emotional mimicry and intention, as well as their interconnection."

Point 2: Line 524. Keep the same outline as the previous quotations.

Response 2: We sincerely value your suggestions and have made corresponding adjustments. The updated version now states, " Nonetheless, this study does have some limitations. For instance, we categorized happiness as an affiliative emotion and anger as a distancing emotion based on the social function of emotions [5].”

Point 3: Check lack of space line 475.

Response 3: We sincerely value your suggestions and have made corresponding adjustments. The updated version now states, “While previous studies have utilized EEG to investigate emotional mimicry [37,38] they have not encompassed the conditions of integrating facial emotions and context.”

Point 4: Check Reference 5. It is incomplete.

Response 4: We sincerely value your suggestions and have made corresponding adjustments. The new version now states, “Fischer, A.; Manstead, A. Social Functions of Emotion and Emotion Regulation. In Handbook of emotions; Lewis, I., Haviland-Jones, J., Barrett, L., Eds.; New York: Guilford, 2016; pp. 456–468.”